# Disruptive Innovation in the Context of Retailing: Digital Trends and the Internationalization of the Yiwu Commodity Market

Wan Liu [1,*] and Steven Si [1,2]

1   School of Management, Zhejiang University, Hangzhou 310058, China; ssi@bloomu.edu
2   Zeigler College of Business, Bloomsburg University of Pennsylvania, Bloomsburg, PA 17815, USA
*   Correspondence: liuwan@zju.edu.cn

**Abstract:** The prevalence of disruptive innovation practices, enabled by the advancement of digital technologies, has greatly changed the way SMEs innovate and the competitive landscape of today's retail industry. This study seeks to understand how disruptive innovation has been adopted for the purpose of internationalization across retailing SMEs in Yiwu's Commodity Market. To answer the research questions, the approach used in this study utilized a qualitative research approach in combination with semi-structured interviews. In this way, the chronology of several phases of Yiwu's Commodity Market's development into a global market center is presented, based on analysis of the data. The findings of this study provide an insight into how to facilitate disruptive paths to achieve the internationalization of SMEs through dynamism of the market, strategy positioning, and capability construction. This study contributes to literature on disruptive innovation by providing and testing a model of internationalization mechanisms that SMEs can use to coordinate digital disruptive innovation-related activities. The study also provides insights for policymakers and SMEs in the retail industry about the importance of digital technologies for motivating potential entrepreneurs to pursue new ventures.

**Keywords:** digital technologies; SMEs; disruptive innovation; internationalization; Yiwu market

## 1. Introduction

For many years, Chinese retail SMEs have been at a disadvantage when participating in global competition [1]. Retail enterprises in emerging markets tend to have major latecomer disadvantages; to be less market-orientated, to have started late, and to be at a stage of development lagging far behind those of developed countries [2]. Generally, Chinese SMEs also have low innovation consciousness, poor brand-building awareness, and lack the key technologies needed to implement disruptive innovation. Since China's reforming and opening up, the main way Chinese enterprises have participated in the international division of labor linked to product retailing has been through the introduction of foreign capital or actively accepting outsourcing orders from developed countries' enterprises.

In this way, Chinese retail enterprises have been latecomers at the low end of the global value chain, and their freedom for development seriously squeezed [3]. Chinese SMEs have been forced to depend on multinational companies or overseas importers and exporters, with the multinational companies occupying the upper end of the global value chain by virtue of their established brands and use of advanced technology [4]. It is easy to place late-developing enterprises in the undesirable pattern of being caught in 'impoverished growth' [5]. In addition, enterprises which have been involved in producing labor-intensive products for a long time are easily captured by the more advanced 'high-end' companies and, thus, become locked long-term at the low end of the value chain [6]. Therefore, for China's small and medium-sized retail enterprises, the direction of future development should be to transcend the roles of the international incumbents, to vigorously develop

their knowledge-intensive productive service industries, to cultivate their competitive advantages, and to achieve climbing up to the high end of the value chain [7,8].

Kim et al. (2020) emphasized that 'SMEs in emerging market countries have special advantages, and their disruptive innovation activities are yet to be examined' [9]. Research carried out on Chinese firms emphasizes that disruptive innovation is an important internationalization strategy that can be valuable in providing room for both the growth of the technological and organizational capabilities of local firms, and in securing the needs of linked local populations [10,11]. Research on disruptive innovation theory also points out that the combination of the new technology and market tracks can provide a 'window of opportunity' for latecomers to catch up [12].

In this paper, we attempt to apply this theoretical framework to chronologically examine these links in the case of SMEs from the retail industry, through different stages of the disruption innovation process, a line of research proposed by scholars but not yet developed [13]. For the competitiveness of small businesses, it is especially important to learn how these three strategic activities relate to each other and how they may be fostered, keeping in mind that small companies are often more flexible than their larger counterparts in terms of disruptive activities and digital technology adoption. Moreover, it is crucial for the national economy, since small firms are a major source of employment, innovation, and growth [14].

In this study, the case of the Yiwu commodity market has been analyzed, the retailing internationalization process of which has been a global breakthrough event. An analysis of disruptive innovation in the Chinese retailing industry that took place between the 1980s to the present is made, key factors such as market dynamism, logic orientation, and the construction of capabilities being examined. In addition, an analysis is made of the ways in which the internal technology paradigm and external networking have facilitated the development of a complex disruptive innovation system, despite the existence of a predominant poverty-reduction culture [15], a low level of training in digital technology, and the industry's under-matured state. This approach is considered useful in explaining the evolution of disruptive innovation diffusion, which has overcome inhibiting barriers affecting the retail industry worldwide.

The paper is structured as follows: The paper begins with a description of the research methodology, then introduces the reader to the context of the retailing industry. Next comes an analysis of the theory background to disruptive innovation and SMEs' internationalization. A case study is then discussed. The paper concludes by discussing the findings and their implications. Based on this, this paper proposes the path of internationalization through disruptive innovation at different historical stages of SME development through a case study and validates it with the case of YiWu City's Commodity Market, Zhejiang Province. By analyzing the capability requirements of value network construction at different stages of disruptive innovation, the paper aims to provide an effective approach for new start-ups to achieve internationalization through disruptive innovation.

## 2. The Theoretical Background

### 2.1. Disruptive Innovation

The topic of disruptive innovation has received much attention from practitioners and academics alike [16–22]. It describes a type of innovation in which a specific process occurs in which incumbents are eventually displaced by newcomers [23]. Disruptive technologies, as defined by Christensen and Bower (1996) in their key article on the disk drive market, are innovative technologies that introduce new performance criteria that suit emergent customers but underperform on existing features that satisfy mainstream customers [24]. Over time, disruptive innovations improve on the traits desired by the mainstream market, in this way infiltrating each market niche from the bottom upwards [25].

Disruptive innovation theory is commonly applied in current research on incumbent SME's rivalry. Disruptive innovation, as opposed to 'sustaining innovation' [17], refers to innovations that establish new markets and value networks, eventually disrupting older

system models. The concept is used by Christensen to explain why successful companies have sometimes failed to adopt new technologies and business models, allowing disruptive innovators to gain competitiveness. In contrast, 'sustaining innovation' maintains the improvement of a product's functions, based on the original innovation [26,27]. A simple example of sustaining innovation is Apple's continuous release of the iPhone series, from iPhone 1 to the new version, iPhone 13.

Christensen and Raynor (2003) also distinguish between low-end disruptive innovations and high-end disruptive innovations (new markets) [23]. The former refers to those products and services that provide inferior performance at a lower price, having no other performance improvement, while the latter is described as being products and services that provide better performance in different attributes for mainstream customers [28]. The differences between disruptive innovation and sustaining innovation are listed in Table 1.

**Table 1.** A comparison between sustaining innovation and disruptive innovation.

| | Sustaining Innovation | Low-End Disruptive Innovation | New-Market Disruptive Innovation |
|---|---|---|---|
| Definition | Continuous improvement of the performance of the original product according to customer needs, and its technological progress generally exceeds the market demand. | Disruptive innovation may deviate from the mainstream market by introducing a product or service that meets the needs of low-end users. | Creating new demand for a new technology, resulting in consumers demanding this new product. |
| Target market | Mainstream market | Low-end market | New market |
| Target products | Mainstream products Function improvement | Low-end products 'Good enough' in function | New products Easy to use |
| Type of diffusion | Monopoly penetration | Low price encroachment | Pre-emptive penetration |

Companies that successfully carry out disruptive innovation have a strong approach to explaining and addressing the needs expressed by market niches or new market segments. Therefore, the challenges that existing enterprises need to overcome in developing and responding to disruptive innovation, involve developing the ability to predict market trends and attitudes, and to 'control' the trajectories of new technologies [29,30].

Despite its widespread use as a change theory, academics and practitioners have questioned as somewhat simplistic disruptive innovation's role in technological transitions as being either disrupting or being disrupted. As an example, it can be challenging to use the concept of disruptive innovation when applied to complex social systems.

*2.2. Disruptive Innovation as an Effective Way to Promote Competitive Advantage for Latecomers*

In the process of catching up with incumbent enterprises, latecomers identify opportunities for disruptive innovation by fully integrating various network resources and engaging in continuous organizational learning through establishing vertical and horizontal links with other enterprises and organizations [31,32].

The basic characteristics of disruptive innovation are characterized by enterprises that start from low-end markets, fully utilizing their initial advantages as their base, and which have customer needs at their core, thereby gaining market advantage by finding ways to destroy old markets or create new ones [21,33]. The construction and evolution of disruptive innovation for the competitive advantage of latecomer enterprises are mainly carried out in the following ways:

First, disruptive product innovation. Christensen's research found that incumbent firms tend not to pay much attention to the development of disruptive technologies due to factors such as product preferences and superior market performance in the mainstream market, and some firms that cannot compete with the mainstream products bring disruptive products to market to avoid monopoly blockage by incumbent firms [34]. Therefore,

when the disruptive products are in the introduction and growth periods, there are more competitors grabbing the limited market share [35]. In the growth period, with the further improvement of the disruptive products and through effective cultivation of consumer preferences, the disruptive products will grab the market of the mainstream products; although the mainstream products of the incumbent companies have excellent performance, they have to face the risk of market shrinkage and consumer shift [36]. Taking advantage of their first-mover advantage, the incumbent needs to solve realistic problems when formulating product strategies, such as when to launch disruptive products, what kind of disruptive products to launch, and what strategies to launch disruptive products with [36,37].

Second, disruptive technology innovation. A technology, by latecomers, that changes the basis of market competition by increasing the performance of existing products. From a technical point of view, such disruptive technology has a different 'performance trajectory' from the mainstream market technology but is not a complete overthrow of previous technology and is not particularly difficult to achieve, which is especially suitable for latecomers with few technical resources [38]. As an example, assembling components manufactured by existing technologies according to a completely new product structure can provide new functional attributes for customers. Such new performance attributes can be improved extremely quickly, allowing this new technology to subsequently penetrate the mainstream market [39]. As a result, this technology will be needed by mainstream customers. It may be too late for the incumbent company or supplier to develop the new technology; the pioneers of the new technology have already captured the market.

Third, disruptive new market innovation. For a latecomer with very poor technological resources, it may be very difficult to implement disruptive technological innovation, so it can consider starting in a new area of the market, using its experience and resources in the global industrial division of labor and reliance on its home country and local market, to break existing attitudes and redefine customer value, product, or service in new delivery paths [9,39]. Disruptive new market innovation may result from breaking the old rules of competition, or a reconfiguration and integration of elements of the traditional value chain, to achieve the effective allocation and use of innovative elements and the creation of a new market [40].

### 2.3. Research Gaps and Research Questions

Latecomers can shorten the gap with incumbents through disruptive innovation, but can market innovation create disruptive opportunities for emerging market retail firms that possess inferior technology bases and innovation resources? If so, what path of disruptive innovation should be followed? Through literature analysis, this paper finds that existing studies have two major research gaps: first, they focus on the catch-up performance of latecomer SMEs, and lack research on the specific innovation strategies that latecomer SMEs should adopt in the process of catching up; second, they focus on the disruptive innovation of latecomer SMEs in developed markets and lack research on the specific innovation strategies of latecomer enterprises in emerging economies.

This paper argues that the three realization paths of disruptive innovation for latecomer firms proposed by existing studies are feasible, but the explanation of the specific internationalization mechanism of the three paths is not strong enough. Based on this, this paper proposes to address two questions from the perspective of disruptive innovation: why Yiwu can continue to generate disruptive innovations and how to understand the uniqueness of the internationalization paths of Yiwu SMEs.

## 3. Method

### 3.1. Overall Approach and Context

The main focus of this article is a description of how disruptive innovation has been adopted for the purpose of internationalization across an entire industry. It also explains how digital technologies have affected competitive dynamics. In order to do so, an industry where digital technologies have already been in use on large scale was targeted. The Yiwu

retail industry provided a compelling case as almost the entire industry had transitioned to online e-commerce during the period of the 1980s to the present.

Yiwu (a city in Zhejing province) is famous for its small commodity market. It is an international commodity distribution center and an important base in China for foreign merchants to purchase small commodities [41]. With active disruptive innovation activities and the adoption of digital technology, the retail industry has become one of the leading industries in Yiwu [15]. This industrial economy has been rapidly aggregating and strengthening, and a group of predominant small commodity markets has formed, such as markets in socks, zippers, cosmetics, and others. Compared with other Chinese traditional industries, the Yiwu SME retail industry has used more active disruptive innovation activities and has achieved better performance in technology innovation. By using an approach combining theory and studying practices used, this paper has investigated enterprises' disruptive innovation behavior in the Yiwu retail industry, and a new explanation of how to solve the difficulty of insufficient incentive for disruptive technology in the retail industry is advanced. Several other aspects of this industry are also still changing, including online display and trading, smart warehousing and logistics, the development of convenient customs clearance systems, global supply chain services, credit data collection and application, and financial empowerment. The Yiwu commodity market is currently well consolidated and achieved 218.79 billion yuan in network retail sales in 2021, ranking first among counties and urban areas in Zhejiang, thus making it a suitable choice as a representative of the Chinese retail industry [42].

To address the proposed purpose, we investigated different phases of disruptive innovation in the retail industry in Yiwu, China. In particular, we observed that there were several reasons why the traditional retail industry, centered on physical shops, changed disruptively in order to place itself in a better position for purchasing internationalization. According to Zaki (2019), the entire retail industry is now due to go through disruptive changes. The time would seem opportune to institute the next generation of value creation in digital transformation, based on new participating actors and changes to existing ones, involving the incorporation of new technology and digital components [43].

*3.2. Data Collection*

This paper focuses on the questions of 'why Yiwu can continue to generate disruptive innovations' and 'how to understand the uniqueness of the internationalization paths of Yiwu SMEs'. These 'why' and 'how' type questions, where the direction is clear but the conclusion is not yet clear, are suitable for an inductive case study approach [33]. This approach is useful when existing theories do not answer existing questions, when the question involves a process or evolves over time. In order to ensure a complete chain of data evidence and thorough case analysis, the research in this paper used semi-structured interviews and has tried to find the evidence behind the successful exploration of disruptive innovation by analyzing the specific practices of typical SMEs in Yiwu [44,45]. As a pioneer in reducing poverty and promoting prosperity in China, Yiwu is a model for other disruptive innovation frontiers and internationalization around the world [46]. Analyzing and summarizing the experiences of Yiwu's enterprises in their inspirational development of disruptive innovations is of high value [15]. Participants in the semi-structured interviews were individuals with interpretative and process knowledge in an area of expertise. We chose local entrepreneurs and policymakers who could provide us with insights from disruptive innovation projects, as well as first-hand knowledge of the issues under investigation. We have examined our data and found discrepancies across expert opinions using the approach we chose [47]. Second, the professionals chosen are known to have a thorough understanding of SME innovative technological applications in Yiwu. They come from a variety of backgrounds: state-owned enterprise managers, government officers, SME entrepreneurs, and business professors from Zhejiang Province, but they all have a reputation for having a thorough understanding of Yiwu disruptive

innovation processes, the roles of SME entrepreneurs in the Yiwu small commodity market, and the history of Yiwu's economic development in recent years [15].

In total, 65 participants involved with the Yiwu retail market were interviewed in December 2021. The participants had an average age of 46 years, with two to ten years of experience working or investigating the Yiwu commodity market over the past 30 years. Interviews were recorded and transcribed. In addition to the conducted interviews, an extensive amount of secondary data was collected. This data include annual reports from publicly listed retail companies, marketing material such as case studies, and newspapers about the Yiwu retail market's development. Data about market share, wholesalers, and other industry statistics along with firm specific data in the empirical section came mostly from these sources. The case description below emerged from combining interview material and that from secondary sources. The data in this paper are mainly from semi-structured interviews with respondents, supplemented by on-site observations and secondary data collection. Based on the diversity of the data sources, the data has been triangulated to ensure its reliability [48]. The data collection methods are displayed in Table 2.

**Table 2.** Data collection.

| | Data Sources | Data Information | | |
|---|---|---|---|---|
| Data collection sources of Yiwu's case | Semi-structured interview | Interview time length | Interview word count | Target interviewees |
| | | 15 h | 800,000 words | 65 people |
| | On-site observation | Visiting Yiwu Commodity City and Yiwu Cultural Exhibition Hall | | |
| | Second source data collection | Annual reports from publicly listed retail companies, marketing material such as case studies and newspapers about Yiwu retail market development. | | |

*3.3. Data Analysis*

Drawing on Miles and Huberman's coding suggestions, this paper divides the coding effort into three stages, following the conventional coding steps [49]. First is the data simplification stage, in which the research team sifts and refines the original textual material to transform the complex original textual material into an easily recognizable coding table [50]. The data transformation process went through two main steps:

The first step was to define the initial codes, i.e., to identify the sources of the different data, e.g., labeling interviews from company founders and executives as M1; interviews from department heads or employee representatives as M2; data from public sources as S1; and data from internal documents as S2. The second step was to refine the coding concepts, which required a clear understanding of the meaning of the concepts related to the study topic based on the literature analysis. The coding concepts in this paper include the inception of disruptive innovation, low-end market disruptive innovation, and new market disruptive innovation, as shown in Table 3.

**Table 3.** Connotation and measurement of constructs.

| Phases of Yiwu's Development Path | Time of Yiwu Commodity Market Development | Sub-Variables | Predominant Characteristics |
|---|---|---|---|
| Sustaining innovation | 1980s–1990s Yiwu local commodity market | Low dynamism of the market | Relatively unmature market; Clear user needs from local people; Stable orders |
| | | Imitation positioning | Pursuit of products' low prices; Resource bricolage |
| | | Capabilities accumulation | Accumulation of prior knowledge; Continuation of organizational practices |

**Table 3.** *Cont.*

| Phases of Yiwu's Development Path | Time of Yiwu Commodity Market Development | Sub-Variables | Predominant Characteristics |
|---|---|---|---|
| Low-end disruptive innovation | 1990s–2001 Yiwu national commodity market | Middle dynamism of the market | Differentiation of user needs; Market dynamism changing |
| | | New market positioning | Restructuring core resources Shifting and laying out new markets |
| | | Capabilities transformation | Continuous acquisition of new knowledge Abandoning old organizational practices |
| New-market disruptive innovation | 2001–present Yiwu International commodity market | High dynamism of the market | Industry disruption New needs from the cutting edge markets' customers |
| | | Digital technology positioning | Strengthened R&D department Recruiting senior mechanics Seeking a joint venture partner |
| | | Capability substitution | New and old knowledge turnover Formation of new organizational practices |

## 4. Case Study Analysis and Results

*4.1. Overall Approach and Context: An Overview of Yiwu's History towards Internationalization*

Our data suggest that the Yiwu commodity market developed its platform in five stages, bridged by three transition phases. While the interactions with the platform and its participants were similar within each phase, they differed between the stages. Overall, the interaction between the platform and its participants changed throughout the phases, enabling the platform to move from inception to the growth toward market disruption. (1) The first and second stages were characterized by the inception of disruptive innovation—'feather for sugar' and stall entrepreneurship. During these periods, Yiwu peasant entrepreneurs orchestrated the internal resources and were closely affiliating with their neighborhood [51]. The peasants used innovative approaches including disruptive innovation to overcome difficulties in order to undertake diversified entrepreneurial activities. (2) In the third stage, which overlapped with the first two stages, Yiwu was growing into a local small commodity trading platform, exploiting knowledge, differentiating from the parent and other dominant competitors, and aiming to build legitimacy for a low-end products category. Based on the identification of value beyond the existing platform, the third phase overlapped with the first two stages. (3) In the fourth stage, Yiwu became a small national commodity market. During this stage, Yiwu steadily built its solid infrastructure for retail business nationally and started to implement new digital technology imported from abroad. This stage was characterized as disruptive technology innovation. (4) In the final stage, Yiwu was growing into an international trading market. Thereby, Yiwu was exploiting the platform's underlying digital architecture, while benefitting from its previously established distinctiveness and the platform's existing network of potential customers [52]. We have collected details of disruptive events in Yiwu's path to becoming a global market town, in Table 4. Figure 1 provides the timeline of Yiwu's disruptive path towards internationalization.

Table 4. The disruptive events of Yiwu commodity market's internationalization.

| Time | Disruptive Events Description |
|---|---|
| 1970s–1980s | 'Feather for sugar' to stall entrepreneurship |
| 1982 | The inception of Yiwu local commodity market: Yiwu County Party Committee and County Government allow farmers to engage in business and approve the opening of urban and rural markets |
| 1984 | Yiwu County Party Committee and County Government formally propose the development strategy of 'building a county for business'. |
| 1986 | The turnover of Yiwu local market exceeds 100 million yuan, the catchment area for Yiwu local market grows from the surrounding counties and cities and extends into the province and beyond. |
| 1993 | The establishment of China Commodity City Group Co. China Commodity City Group Co., Ltd. (Yiwu, China)with 40% government ownership. |
| 2001 | The first and second phases of the International Trade City market are built, and the upgrading of the China Commodity City into a modern international commodity trading platform. |
| 2002 | On May 9, 2002, China Commodity City Group Co., Ltd.'s shares are listed on the Shanghai Stock Exchange with the stock code 600415. |
| 2006 | 'The "Yiwu-China Commodity Index' is officially released on 22 October 2006. |
| 2013 | Since 2013, Yiwu online transactions have outperformed physical transactions in terms of sales. |
| 2017 | In 2017, Yiwu's e-commerce transactions amount to 222 billion yuan. |
| 2018 | Yiwu introduces 1854 kinds of imported goods from 121 countries and regions around the world, up 35.23% year-on-year. |

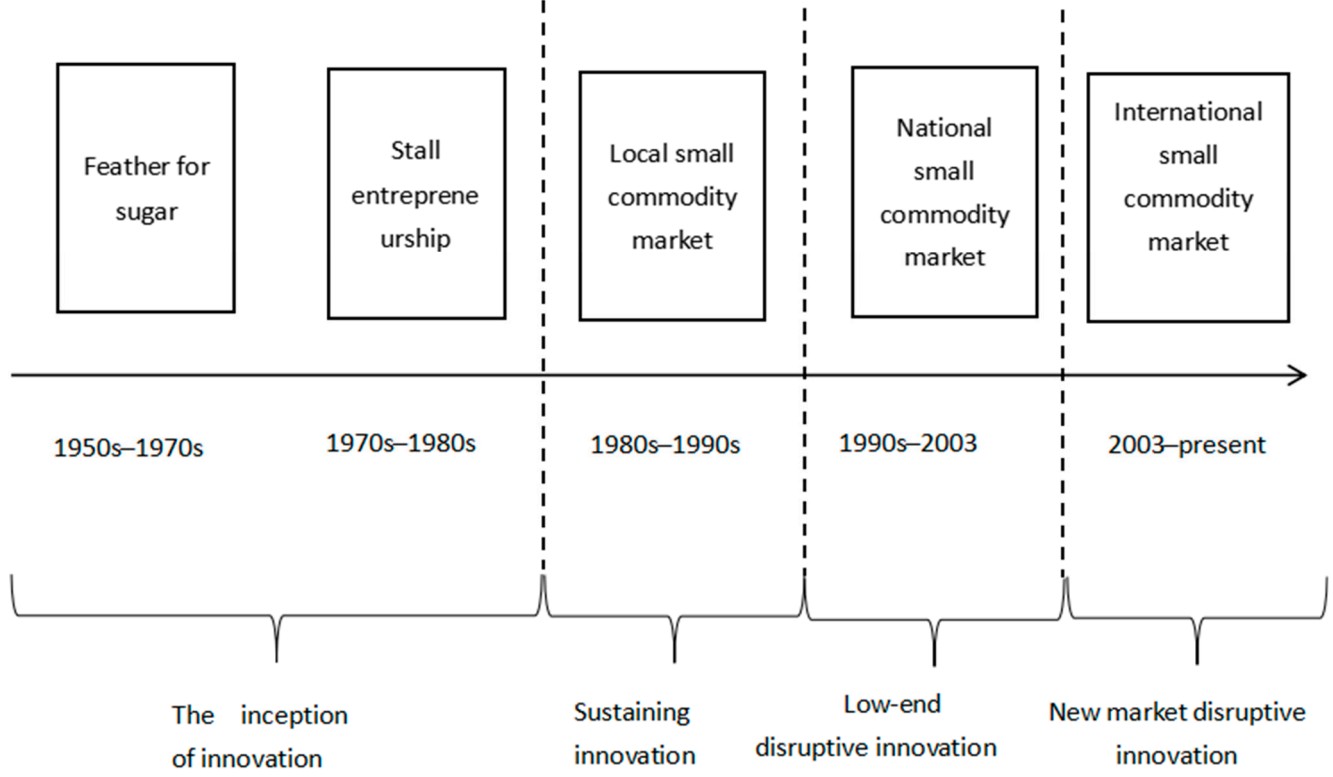

Figure 1. Timeline of Yiwu's disruptive path towards internationalization.

4.1.1. Stage 1: Feather for Sugar

More than 30 years ago, Yiwu was still seen as representative of the extremely poor areas of Zhejiang Province. At that time, most Chinese farmers lived only in rural areas and cultivated the fields allocated to households. However, the mountainous area where

Yiwu is located had even worse conditions, being not along the border or near the sea, with barren land and poor people, and a lack of natural resources [15,53]. Additionally, its terrain was not suitable for agricultural development. With limited resources and restrained external support and assistance, farmers were forced to find other ways to increase their income to make ends meet. A typical example was trading goods through 'chicken feathers for sugar', where people exchanged brown sugar, straw paper, and other inexpensive items for chicken feathers and other waste products from residents' homes for a small profit [15].

### 4.1.2. Stage 2: Stall Entrepreneurship

In this stage, the 'chicken feather for sugar' trade was still considered illegal. Nevertheless, farmers in Yiwu overcame the difficulties and engaged in various entrepreneurial activities through innovative activities. In 1981, the central government's Document No. 1 officially allowed farmers to participate in the 'chicken feather for sugar' trade. The central government officially allowed farmers to participate in various business activities, so farmers in Yiwu Chucheng and Lane 23 seized the opportunity to set up stalls on the town's roadside [15,53].

In September 1982, Yiwu county government officially opened a small department store market in Houcheng town. At that time, 9000 yuan was invested to promote 700 places for the open-air market, cement boards being laid [54]. The turnover of the small commodity market in that year was 0.92 million yuan. In 1983, the number of stalls in Huqingmen market increased to 1050, with a turnover of 1.44 million yuan. In 1984, the turnover of this market reached 2.31 million yuan, and the commodities were mainly sold to the counties and cities around Yiwu. With the support of the CPC Central Committee, Huqingmen market was allowed to exist and develop. During the 12th National Congress, under the direct leadership of the County Administration for Industry and Commerce, and with the cooperation of relevant departments, the Huqingmen small department store market was modernized and took on a new look. After the market was officially recognized, its speed of development exceeded anyone's expectations.

### 4.1.3. Stage 3: Local Small Commodity Market

This phase was mainly marked by the establishment of small commodity markets at individual and organizational levels from 1982 to 1990. The construction of the third generation of China Commodity City market started in November 1985. The site is located in the plot east of Chengzhong Road (then called Huancheng Road) and between Qianda Road and the standard parts factory. The total investment was 4.4 million yuan. It was completed and opened in 1986, with 4096 fixed stalls, covering an area of 44,000 square meters. There are comprehensive commercial services and management service buildings for industry and commerce, taxation, posts and telecommunications, and finance [55]. After several more expansions, by the end of 1990, the third time construction phase of the China small commodity city resulted in the largest professional wholesale market for small commodities in China, covering an area of 57,000 square meters, with 8503 fixed stalls and more than 1500 temporary stalls. In 1991, the turnover of the small commodity market reached 1.033 billion yuan, breaking the 1 billion goal for the first time. It was at this time that the status and role of the small commodity market began to be recognized by an increasing number of people, and many grass-root entrepreneurs participated in the market initiative.

### 4.1.4. Stage 4: National Small Commodity Market

The fourth generation of China's small commodity city market was started in 1991 and put into use in 1992, with a total of 7100 stalls. In March, Yiwu small commodity market ranked the list of the country's top ten markets first released by the State Administration for Industry and commerce [56]. Yiwu Co., Ltd. (Yiwu, China), the predecessor of Yiwu Co., Ltd., was renamed 'China small commodity market' in August 1993. On 4 June 1994, the phase II project of the fourth generation of China Commodity City market passed the

handover acceptance. So far, the construction area of the commodity city has expanded to 228,000 square meters, and the number of stalls has increased to 23,000. In 1995, the turnover of China's small commodity cities reached 15.2 billion yuan.

### 4.1.5. Stage 5: Yiwu Market as an International Market Town

By the end of the 1990s, the successful story of Yiwu Market had encouraged other regional authorities in PRC to follow suit, inducing intense competitions. At the same time, the fast development of e-commerce posed remarkable challenges to the very foundation of Yiwu market as well [48]. Its suppliers face decreasing profits owing to fiercer domestic competitions for consumers from other alike specialized markets and challenges from e-commerce platforms like Taobao. The Yiwu authorities took actions to respond to those challenging difficulties via increasing the global presence of Yiwu and taking advantage of the development of EC.

In 2002, they rolled out a novel developmental plan to establish a worldwide commercial hub. Such a novel plan intended to form a novel commercial pattern by the integration of global trade fairs, overseas trading service, logistic service, product inspections, the customs declaration and global financial transactions in one unified market. The initiation of the Yiwu International Trade Fair in 2001 was a vital part of such novel strategy. By 2006, 103,205 merchants had been attracted to the yearly Yiwu Fair, of whom 16,056 were foreigners from 161 nations. Such a fair has become the 3rd biggest trade fair nowadays in China, after the China Import and Export Fair in Guangdong and the East China Fair in Shanghai. Inspired by the successful story of such trade fair, various novel fairs have been launched in Yiwu as well, with the quantity increasing from 11 in 2001 to 27 in 2005.

In 2005, the overall trading volume of those fairs reached 10 billion RMB, taking up more than 1/3 of the overall trading volume of Yiwu market. The Yiwu authorities invested 200 million yuan to establish a new Yiwu International Logistics Center, which provides one-stop clearance for export and import commodities. With a yearly tonnage of 400,000 TEUs, this center has become a vital inland port for the cities nearby. In 2007, export products took up 70% of the overall trading volume of Yiwu Market. Over 10,000 buyers from more than 100 nations live in Yiwu due to its convenient commercial environment. Moreover, over 40% of airline passengers that enter and leave Yiwu are foreigners. Transnational passengers that enter and leave Yiwu by planes surpassed 500,000 in 2007.

The prosperous EC development across the globe brings both challenges and chances for Yiwu. The convenience of EC platforms decreases the dependence on physical stores. The development of business-to-business (B2B) corporations like Alibaba has attenuated the importance of commercial hubs featuring physical stores. The advantages of a conglomerated market in a certain place are weakened nowadays, as consumers can contrast various commodities across the nation conveniently on the Internet. Nevertheless, EC can help physical stores reach more potential customers restricted by spatiotemporal factors in the past. Yiwu Market's international goals nowadays are based on the EC development in PRC and the first-mover advantages in business competitions. Yiwu Buy was initiated in 2012 as a comprehensive on-line transaction platform for vendors sourcing their commodities from Yiwu [42]. The thriving and intricate EC enables global consumers to buy from the Market and from international merchants and suppliers in Yiwu, which allows them to compete against major retailers like Walmart, Carrefour, and Tesco. These commercial giants have become dominant by purchasing commodities straightly from PRC [57]. Unlike Taobao, which offers an on-line platform for small vendors and retailers to open on-line stores (business to commerce [B2C] or commerce to commerce [C2C]), Yiwu Gou serves suppliers that operate in Yiwu Market by offering them an online platform to sell commodities to retailers, like the online stores active on Taobao (B2R).

The Yiwu Market was featured by small products at the beginning and evolved into the biggest wholesale market of small products across the globe. Nowadays, with the initiation of the Yiwu International Trade Comprehensive Reform Pilot Zone and the Yiwu Crossborder EC Comprehensive Pilot Zone, Yiwu will be the first station for international

commodities to enter PRC. It will be a vital logistic center for global commodities, and a vital base for cross-border EC, creating a center featuring 'buy globally, sell globally' [53].

### 4.2. A Way to Global Market Town: 3 Phases of Yiwu's Disruptive Innovation Path

In this paper, we have summarized and analyzed the data of Yiwu's retail industry and find that Yiwu's retail industry has achieved disruption of the existing market at different phases of development. In the process of achieving disruption, SMEs in Yiwu Commodity City have fully utilized and tapped into the dynamism of the market, used strategy positioning and capability construction, and have experienced a long period of time in which they have achieved disruption of the existing market. Combining the disruptive features of its chronological development line, this paper divides its disruption process into three phases: Sustaining innovation (1980s–1990s); low-end disruptive innovation (1990s–2003), and new market disruptive innovation (2003–present). Figure 2 presents a conceptual model of Yiwu's disruptive path towards internationalization.

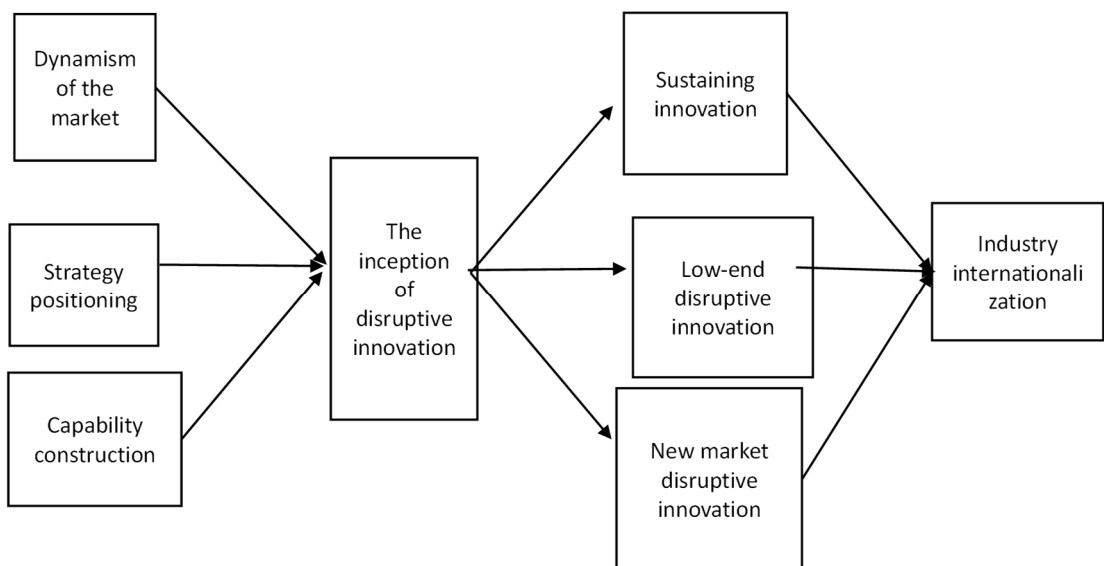

**Figure 2.** Conceptual model of Yiwu's disruptive path towards internationalization.

#### 4.2.1. Phase 1: Sustaining Innovation (1980s–1990s)

Low market activity. In Yiwu, the land and other resources were remarkably limited, and rural people merely relied on agriculture, hence county governments granted greater leniency to rules pertaining to transactions and barters. Moreover, the need for consumption-based commodities was fairly steady in Yiwu, with the primary demand being daily products, because local residents did not feel the alike commercial convenience in contrast to other larger cities [15]. Small booths and stalls emerged in Yiwu, and by the early 1980s, merely a few people participated in those transactions. Consequently, market activity was comparatively low as these transactions were supported by the laws then, which hence posed challenges to financial and other activities.

Product positioning. Despite a lack of regulation, limited resources, and outdated technology, Yiwu's family businesses rapidly expanded production and trade. By producing low-price and good-quality products, the first generation of peasant entrepreneurs gained their reputation in the local market and prospered. In addition, Yiwu's reputation as a trading center was enhanced by a new commodity market, and 'Zhejiang China Commodity City Group' was established in 1982 by the informal business association [51]. The local government, which was also unclear about the direction of national policy, took a calculated risk by allowing farmers in Yiwu to engage in commerce, logistics, competition, and trade. It also set up a special committee to manage the newly formed market [42].

Capabilities accumulation. In this period, the market environment in Yiwu Commodity City was relatively stable. In this period of market stability, each incumbent enterprise established its own market or market segment, i.e., the enterprise could meet the demand of the market or market segment with its own products and could, thus, effectively provide products for its own market [58].

At this time, although there were differences and conflicts of interest between different entrepreneurs, the distribution of market resources among entrepreneurs was symmetrical [59]. As a result, the entrepreneurs mainly used their experience and prior knowledge to look for opportunities to bring into existence new goods and services and market them at a profit.

In such a market structure, the peasant entrepreneurs always used business models that followed existing organizational practice. This period was when the innovation model of Yiwu Commodity City showed characteristics of sustaining innovation. By virtue of their dominant position in the market and the available resources, entrepreneurs at this time were able to earn small profits, enough to make ends meet [15]. Each entrepreneur was eager to find new product segment areas segments to diversify their businesses and increase their profits.

4.2.2. Phase 2: Low-End Disruptive Innovation (1990s–2001)

Middle dynamism of the markets. Compared with the local Small Commodity City stage, in the national Small Commodity City stage, Yiwu faced differentiated user needs, the industry pattern was changing, and market dynamics changed from a low to a medium level of activity. From the mid to late 1990s, China's rapid economic development, and internationalization of professional markets were far from the point of maturity. This state of affairs, combined with the rapid development of China's retail industries, and the low degree of openness of the domestic market, resulted in this period in Yiwu Commodity City's professional market operations being mainly domestic operations [42]. Again, through the disruptive new market innovation business model, Yiwu Commodity City professional market quickly gained a place in the national market, through participation in national division of labor and other business forms.

New market positioning. This period was the initial stage for the exploration of e-commerce [60], which has obvious disruptive innovation characteristics, and most of the application personnel used e-commerce to promote business and release information, including their communication with customers. From the viewpoint of market demand, small commodities are extremely popular, so consumers' demand for sellers were very clear and order volumes relatively stable [61]. From the strategic positioning point of view, the supply of small goods by retailers, was mainly concentrated in the low-end product market, 'fighting the price war' seen as the main way to compete [62]. In addition, Yiwu's small and medium-sized enterprises began to become increasingly conscious that they could not rest in the low-end product market, but should jump out of this 'comfort zone' to seek for longer-term development [63]. Yiwu Commodity City's leading approach at this time was reflected in the phrases 'transfer and layout to new market' and 'adjust the structure of core resources' [64].

Capabilities conversion. Disruptive innovation provides an important way for latecomers to catch up with advanced incumbents [32,65,66]. In the early stage of competition with advanced incumbents, latecomers usually lag behind in technological resources, but as they do not have to go through all the technological development stages, can directly use existing technologies [67,68]. They can build their competitive advantages by taking the low-end disruptive market as the breakthrough point, with customer needs as the core, destroying the old market or a creating new market previously ignored by the advanced incumbents [69]. With their market advantage accumulated in the low-end market, latecomer enterprises can pay attention to technological innovation, gradually transferring from the low-end to the high-end of the market; or, in the process of competition with market incumbents, they may capitalize on first-mover advantage and establish brand

advantage and timely brand extension. In the process of latecomers building their competitive advantage through disruptive innovation, the role of productive services evolves from an initial internal management function to a facilitation function, eventually becoming a strategic element [35].

4.2.3. Phase 3: New Market Disruptive Innovation (2001–Present)

By 2005, the international development of China Yiwu Commodity City professional market entered its fourth stage. Yiwu Commodity City professional market trading platform developing its globalization, and transformation of its international business, with the establishment of international brands and marketing networks.

High dynamism of the market. By participating in the global division of labor and actively utilizing various external network resources, latecomers have gradually accumulated the market knowledge and technological knowledge needed for disruptive innovation [70]. Based on their absorptive capacity and complementary assets, latecomers starting at the lower end of the market gain competitive advantage through disruptive innovations in technology, products, and business models [23,71]. During this period, the competitive advantage of latecomers over advanced incumbents was reflected in the market: they chose to enter the low-end markets ignored by the incumbents by breaking old markets or creating new ones, avoiding direct competition, and expanding the market for their firms and industries.

Digital technology positioning. With the establishment of international brands and marketing networks, the internationalization of Yiwu Commodity City professional market has deepened and adopted the digital business model of 'outward-oriented internationalization' [72]. With the expanding network resources of latecomers, exchanges of digital information and logistics between enterprises are becoming more frequent, and the demand for digital services is constantly increasing [73,74]. For latecomers engaged in disruptive innovation, digital technologies adoption requires them to focus on customer needs, to integrate internal and external digital resources, and to either conduct low-end market activity, or create new markets at a low cost recognized by customers [26,75].

Capabilities substitution. During the reproductive process in times of market stability, retail companies still need technological innovation [76]. However, this technological innovation is based on existing technological paradigms; it is an adaptation, modification, and refinement of existing technologies [77]. Sustaining innovation within a paradigm is a process of technological accumulation [78]. With the iteration of the production cycle and the accumulation of technological knowledge, the technology actually used by enterprises contains two different elements: the first is the initial technology stock when the enterprise introduces a technological paradigm (mainly in machinery and equipment, which can hardly form a competitive advantage since any enterprise can purchase it from the market). The second element is the knowledge accumulated by the enterprise through experience accumulation and incremental innovation in the process of production, which is unique to the enterprise. As production proceeds, the increase of this element improves the enterprise's technology and enhances its technical efficiency, which in turn strengthens its market position. More importantly, due to diseconomies of scale and compression time in the process of technology accumulation [78], this part of the technology is not only scarce but also difficult to imitate, thus helping the firm to gain a lasting competitive advantage. Logistic services play a catalytic role in the disruptive innovation activities of latecomers [79,80]. The construction of the competitive advantage of latecomers depends not only on disruptive innovations and production processes, but increasingly on the full participation of productive services in all dimensions of economic development as providers and disseminators of disruptive innovations in the relevant economic sectors.

It can be seen from Table 3 that Yiwu Commodity City has realized three capability reconstruction mechanisms, these being capability accumulation, capability conversion, and capability substitution, based on three dominant logics: dynamism of the market, market positioning, capability reconstruction, respectively, in which the three different levels of

market dynamics low, medium, and high are being followed. The theoretical model is based on the framework of 'disruptive innovation' development's chronological line, which shows the complexity of the internationalization process of late-developing enterprises in Yiwu Small Commodity City. It is worth noting that although the degree of market dynamics, strategy positioning, and capability reconfiguration mechanism of latecomer enterprises in the three stages are different, there is still a common pattern; market dynamics provides the external background premise for capability reconfiguration of latecomer enterprises, while dominant logic is the internal cognitive basis for capability reconfiguration of latecomer enterprises [81]. By continuously promoting the evolution of capability reconfiguration mechanism evolution, latecomer enterprises in Yiwu retail industry have finally achieved the goal of technological catch-up and international development.

## 5. Discussion and Conclusions

### 5.1. Main Findings

Disruptive innovation strategies of emerging market companies are increasingly challenging. Drawing on a qualitative case study of Yiwu retail industry, the biggest commodity market in China, this paper explores how disruptive innovation strategies have enabled a resource-constrained emerging market region to deliver affordable, innovative, and good-quality products with minimal capital investment. The findings of this study illustrate how multiple forms of disruptive innovation can be implemented and managed at organizational and industrial levels. They show how different disruptive innovation activities relate to different stages of new product development processes, and also highlight the complementarities between digital technologies and new market exploration activities [26]. This paper argues that disruptive innovation, which is typically regarded as a behavioral trait or skill that allows entrepreneurs and innovators to operate in challenging environments, can also be a carefully planned and executed strategy conducive to innovation. Thus, it suggests that latecomer disruptive innovators need to position themselves according to the characteristics of the mainstream consumer market, avoiding it and reducing the likelihood of incumbent threats by managing non-visibility.

### 5.2. Contributions

We believe our work makes contributions to several streams of research. First and foremost, we have contributed to the emerging literature on disruptive innovation in emerging economies, by providing theoretical rigor to a stream that has been largely based upon a strong applied focus in developed economies. Second, we have responded to calls for the convergence of literature dealing with disruptive innovation and internationalization by providing a theoretical integration for research at the intersection of the two. Finally, we have contributed to strategy research by systematically identifying key strategic factors that SMEs will have to address to develop disruptive innovation in the different phases leading towards internationalization.

### 5.3. Limitations and Future Studies

This paper also has some recognized limitations. We need to conduct more research in the future to cover gaps in our findings which have come to light as a result of our work.

First, this paper has focused on the windows of opportunity for disruptive innovation brought about by different stages of the market's development and emphasizes that future comparative studies should be conducted on the windows of opportunity caused by the institutional context. Being based on a single case study, we believe the conclusions of this paper should be tested by multiple case studies and large-sample empirical evidence in the future. Secondly, this study is based on a single industry, and to ensure that the research results of this paper are equally feasible theoretically and practically for other industries, the follow-up study should further expand the sample selection and corroborate and expand the findings at a deeper level. Third, this paper has mainly investigated the three dominant competencies in each disruptive innovation period; future research may well be further

enriched and expanded by considering whether the interactions of additional competencies contribute to successful disruptive innovation activities.

**Author Contributions:** Conceptualization, W.L.; Investigation, W.L.; Methodology, S.S.; Project administration, W.L.; Writing—review & editing, W.L. All authors have read and agreed to the published version of the manuscript.

**Funding:** This research received no external funding.

**Conflicts of Interest:** The authors declare no conflict of interest.

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
