# Peer review of "Disruptive Innovation in the Context of Retailing: Digital Trends and the Internationalization of the Yiwu Commodity Market"

_sustainability, doi:10.3390/su14137559_

Round 1

Reviewer 1 Report

The paper is largely descriptive, but I found it to be informative.  I think the only thing remaining is to offer suggestions to the readers for future research.

Author Response

Response: We really appreciate your encouragement and valuable comments. Following your comments, in this revised paper we have included more details to offer suggestions to readers for future studies, please read these at the end section-5.3. Limitations and future studies-in the revised version of the paper. Thank you!

Reviewer 2 Report

The present paper entitled „Disruptive Innovation in the Context of Retailing: Digital Trends and the Internationalization of the Yiwu Commodity Market" starts with the author’s consideration that the major goal of this research are to assessment the causes of disruptive innovation development generated by digital technologies in the traditional retail industry, its spread in the Yiwu commodity market, and the subsequent internationalization in the industry. This paper explores how disruptive innovation strategies have enabled a resource-constrained emerging market region to deliver affordable, innovative, and good-quality products with minimal capital investment. To achieve this goal, the authors have presented a theoretical background analysis based on eighty of references. These references are not only appropriate but also balance major references in the area as well as older and more recent references. I agree with the authors that this study enriches the theory of disruptive innovation and provides management insights for later-stage firms.

Author Response

Response: Thanks for your positive comments. We greatly appreciate your valuable comments and encouragement that help us not only revising this paper but motivating us to work on the ongoing research of late-stage firms. Thank you!

Reviewer 3 Report

Paper needs to improve argument development, formal alterations, grammatical mistakes in this reputed journal.

Major points

  1. The abstract of the paper does not cover all aspects of the paper. There should be one or two sentences on sample/design/methodology in the abstract. Also, use short, simple and concise sentences to make them understandable for the reader. Further, authors have developed their paper from SMEs perspective, but it is not reflected in the abstract.
  2. Authors should discuss the contribution of their study in light of the research gap identified (what previous studies have done, what are their shortcomings and how authors have addressed those shortcomings and contributed to the body of knowledge). Moreover, the claims regarding the choice of Yiwu commodity market and SMEs as a sample should be included with supporting citations. The authors should add a paragraph at the end of the introduction that summarizes the main findings.
  3. It is suggested to cite some more recent empirical studies (from 2020-2022) related to the topic under investigation in the literature review section. Also, provide literature gap at the end and also develop hypothesis statements at the end of the literature.
  4. The choice of the semi-structured interview methodology needs justification. What are the benefits of this method over others? Need detailed discussion on chosen methodology with proper support of references. For method validation, relate underlying assumptions with your analysis.
  5. Results discussion is quite a weak part of the paper although authors have extensively put efforts into the analysis and results section. They should further strengthen their findings with the support of previous literature. How their results are similar and different from literature and contribute to the body of knowledge. Authors are suggested to include some robustness checks (either with a different model or with different measures of the variables) to verify their findings from the main results. It will help strengthen their outcomes and arguments.
  6. Although authors have discussed the findings of the study in conclusion implications of the study should be improved further with a particular focus on the findings of the study.
  7. Also, there are spelling, formatting and grammatical mistakes in the paper. Please carefully go through the paper for formatting errors, spelling and grammatical mistakes before submitting the revised version.

Author Response

Comment 1: The abstract of the paper does not cover all aspects of the paper. There should be one or two sentences on sample/design/methodology in the abstract. Also, use short, simple and concise sentences to make them understandable for the reader. Further, authors have developed their paper from SMEs perspective, but it is not reflected in the abstract.

Response: Thanks for your valuable comments! We have re-written the abstract part which now highlights the topics of sample/design/methodology in keynote sentences, and we also now emphasize the SMEs perspective, according to your comments.

Please see below:

The prevalence of disruptive innovation practices, enabled by the advancement of digital technologies, has greatly changed the way how SMEs innovate and the competitive landscape of today’s retail industry. This study seeks to understand how disruptive innovation has been adopted for the purpose of internationalization across retailing SMEs in Yiwu’s Commodity Market. To answer the research questions, the approach used in this study utilized a qualitative research approach in combination with semi-structured interviews. In this way the chronology of several phases of Yiwu’s Commodity Market’s development into a global market center is presented, based on analysis of the data. The findings of this study provide an insight into how to facilitate disruptive paths to achieve the internationalization of SMEs through dynamism of the market, strategy positioning and capability construction. This study contributes to literature on disruptive innovation by providing and testing a model of internationalization mechanisms that SMEs can use to coordinate digital disruptive innovation-related activities. The study also provides insights for policymakers and SMEs in the retail industry about the importance of digital technologies for motivating potential entrepreneurs to pursue new ventures.

Keywords: digital technologies; SMEs; disruptive innovation; internationalization; Yiwu Market

Comment 2: Authors should discuss the contribution of their study in light of the research gap identified (what previous studies have done, what are their shortcomings and how authors have addressed those shortcomings and contributed to the body of knowledge). Moreover, the claims regarding the choice of Yiwu commodity market and SMEs as a sample should be included with supporting citations. The authors should add a paragraph at the end of the introduction that summarizes the main findings.

Response: Thank you very much for your review. As to your question, first of all, we have added details of the gap in our research in section 5.2.

In addition, we have added more supporting citations and revised this part following your comments and suggestions.

Lastly, we have added the findings of this research at the end of the introduction section. Please see the revised paper.

Comment 3: It is suggested to cite some more recent empirical studies (from 2020-2022) related to the topic under investigation in the literature review section. Also, provide literature gap at the end and also develop hypothesis statements at the end of the literature.

Response: We sincerely appreciate these valuable comments.

First, we have checked the literature carefully and added more recent empirical references related to the topic in the literature review section.

Second, in accordance with your second comment, this time we have provided a literature gap at the end of the literature. Please see the below revised literature review part:

Late comers can shorten the gap with incumbents through disruptive innovation, but knowing how they should achieve disruptive innovation should be a prerequisite for late comers to catch up, or even catch up with incumbents. Through the above literature analysis, this paper finds that existing studies have two major research gaps: first, they focus on the catch-up performance of latecomer enterprises, and lack research on the specific innovation strategies that latecomer enterprises should adopt in the process of catching up; second, they focus on the disruptive innovation of latecomer enterprises mainly in developed markets, and lack research on the specific innovation strategies of latecomer enterprises in developing economies.

This paper argues that the three realization paths of disruptive innovation for latecomer firms proposed by existing studies are feasible, but the explanation of the specific internationalization mechanism of the three paths is not strong enough. Based on this, this paper proposes to address two questions from the perspective of disruptive innovation: first, how do late comer SMEs gain competitiveness in different stages of disruptive innovation? Second, how do late comer SMEs advance the internationalization process under the influence of different disruptive innovation stages matched with innovation strategies?

Lastly, we fully agree with the comment that the research should add some hypothesis to make it more reasonable. We have made some changes. However, in this study we have perhaps focused too much on the role of disruptive innovation in promoting SMEs’ paths to internationalization. Its mainly based on qualitative case studies concluding with a disruptive innovation model of Yiwu retailing industry. We greatly appreciate your comments and suggestions, and we will keep your comments in our minds for our future research on disruptive innovation and internationalization paths. Thank you again!

Comment 4: The choice of the semi-structured interview methodology needs justification. What are the benefits of this method over others? Need detailed discussion on chosen methodology with proper support of references. For method validation, relate underlying assumptions with your analysis.

Response:

In this revised paper, we have added details of the benefits of using a case study approach in combination with semi-structured interview methodology. We have also cited related references to support our argument.

Please see below:

This paper focuses on the questions of "why Yiwu can continue to generate disruptive innovations" and "how to understand the uniqueness of the internationalization paths of Yiwu SMEs". These ‘why’ and ‘how’ type questions, where the direction is clear but the conclusion is not yet clear, are suitable for an inductive case study approach. This approach is useful when existing theories do not answer existing questions, when the question involves a process or evolves over time. In order to ensure a complete chain of data evidence and thorough case analysis, the research in this paper used semi-structured interviews and has tried to find the evidence behind the successful exploration of disruptive innovation by analyzing the specific practices of typical SMEs in Yiwu. As a pioneer in reducing poverty and promoting prosperity in China, Yiwu is a model for other disruptive innovation frontiers and internationalization around the world. Analyzing and summarizing the experiences of Yiwu’s enterprises in their inspirational development of disruptive innovations is of high value.

Comment 5: Results discussion is quite a weak part of the paper although authors have extensively put efforts into the analysis and results section. They should further strengthen their findings with the support of previous literature. How their results are similar and different from literature and contribute to the body of knowledge. Authors are suggested to include some robustness checks (either with a different model or with different measures of the variables) to verify their findings from the main results. It will help strengthen their outcomes and arguments.

Response: We fully understand your concerns. In the current version of the paper, we have largely revised the findings part based on your comments and suggestions. We appreciate your suggestions about adding some robustness checks to verify the main results. As we mention above, in this study we perhaps focused too much on the role of disruptive innovation in developing SMEs’ internationalization paths. Its mainly based on a qualitative case study that use descriptive data analysis. Most of the information in the analysis part is from semi-structured interviews. We have tried our best to avoid subjective problems. We may use this tool for future studies related to this topic.

We hope our revisions satisfy you; we have done our best to revise and improve the theory part of this paper. Thank you for your valuable comments and suggestions.

Comment 6: Although authors have discussed the findings of the study in conclusion implications of the study should be improved further with a particular focus on the findings of the study.

Response: Thank you for your valuable comment. In the previous version of the paper, we did not clarify this issue. In this revision, we have extensively revised the structure and the theoretical basis of the Conclusion section. We have classified the conclusion part into main findings, contributions, and limitations and future directions. Please see the revised paper.

Comment 7: Also, there are spelling, formatting and grammatical mistakes in the paper. Please carefully go through the paper for formatting errors, spelling and grammatical mistakes before submitting the revised version.

Response:

Many thanks for the comments and suggestions. We have revised this paper several times and also had professional help to improve the formatting, spelling, and grammatical errors you mention. In this revised paper, we have revised it several times and also asked a professional to help improving the paper text. We are using an experienced copy editor to edit all of this revised paper before sending it to your journal.

Round 2

Reviewer 3 Report

I am happy with the revised draft and recommended for publication